# Impact of Dynamic Capabilities on Customer Satisfaction through Digital Transformation in the Automotive Sector

**Pablo Martínez de Miguel** [1]**, Carmen De-Pablos-Heredero** [1][ID]**, Jose Luis Montes** [2][ID] **and Antón García** [3,*][ID]

1   Department of Business Organization (Administration, Management and Organization),
    Applied Economics II and Fundamentals of Economic Analysis, Universidad Rey Juan Carlos,
    Paseo de los Artilleros s/n, 28032 Madrid, Spain; p.martinezde@alumnos.urjc.es (P.M.d.M.);
    carmen.depablos@urjc.es (C.D.-P.-H.)
2   Department of Applied Economy I, Universidad Rey Juan Carlos, Paseo de los Artilleros s/n,
    28032 Madrid, Spain; joseluis.montes@urjc.es
3   Animal Science Department, University of Córdoba, Rabanales University Campus, 14071 Córdoba, Spain
*   Correspondence: pa1gamaa@uco.es

**Abstract:** Technology has impacted businesses in different areas, and, consequently, many companies have found it necessary to make changes in their structures and business models to improve customer satisfaction. The objective was to quantify the effect of dynamic capabilities on customer satisfaction, through digital transformation within the automotive sector. A random sample of 42 questionnaires on 127 surveyed industries was collected during the period 2019–2020 in a pre-COVID-19 context. A structural equation model (SEM) in two stages was applied. In the first stage, two reflective models were built. In a second stage, a structural equation model was evaluated. The results obtained in this study showed that the capabilities of sensing, seizing and innovation were suitably grouped in a construct called "Dynamic Capabilities". A positive influence of Dynamic Capabilities on customer satisfaction was found. Therefore, the companies in this industry should focus on developing dynamic capabilities to improve customer satisfaction. Once the opportunities have been identified, managers take advantage of their potential (seizing) to transform and exploit knowledge in the creation, innovation, process improvement, and definition of strategies to combine new knowledge with that already existing. The digital transformation has contributed to identify the real needs for customers, to contact them and solve their problems, as well as offering products and services by anticipating their needs.

**Keywords:** digital transformation; dynamic capabilities; customer satisfaction; automotive industry; structural equation model (SEM)

## 1. Introduction

Automotive and component manufacturing companies form a tandem of recognized prestige in terms of competitiveness and results. The automotive industry has an important multiplier effect in the economy as it maintains clear links to other sectors. It is an important sector for upstream industries such as steel, chemicals, and textiles, as well as downstream industries such as, for example, ICT, repair, and mobility services. Employment—around 13.8 million people work in the EU automotive sector. Manufacturing (direct and indirect) accounts for 3.5 million jobs, sales, and maintenance for 4.5 million, and transport for 5.1 million. From the economic perspective, the turnover generated by the automotive industry represents over 7% of EU GDP [1–3].

This sector is undergoing a profound restructuring and disruptive innovation, aggravated by the COVID-19 pandemic [3]. Apart from this, customers are looking for more energy efficient and environmentally friendly vehicles, mainly hybrid and electric vehicles [4]. Furthermore, COVID-19 is accelerating the digital transformation process. The

development of dynamic capabilities plays an important role in managing the organization's strategy [5–7].

The automotive business, like many others, has been impacted by digital technologies, leading to the need for companies to innovate their business models by developing their dynamic capabilities, understood as the organization's abilities to reconfigure itself according to the demands offered by the changing environment [8]. Through digitization, the company has the opportunity to interact with customers, which has helped in the creation of new business models [9–13].

Digital transformation exposes new ways in which the organization can stay in touch with customers and consumers and thus create value for them [14]. Customers are active entities, who know their needs and know that they have product and service alternatives to satisfy them [15]. Although the success of a company depends on different factors, one of the most important is to increase its competitiveness in the market to achieve customer satisfaction [16].

Digital transformation is a process of reinvention and reengineering of a business to digitize a company [17]. With the emergence of new digital technologies, such as artificial intelligence (AI), Internet of Things (IoT), mobile and social Internet, blockchain, and big data, companies in almost all industries are undertaking multiple initiatives to explore and exploit the benefits of these technologies [18,19]. Meanwhile, society is facing rapid and radical changes due to the maturation of digital technologies and their power to rapidly penetrate markets, while customer demands are increasing and organizations are facing stiffer competition due to globalization [20,21].

The emergence of digital innovations is accelerating and disrupting existing business models by providing opportunities for new services [22]. Based on the automotive industry, major trends such as autonomous cars, connectivity, and car sharing are creating new business models. These are simultaneously giving rise to new competitors in the market, which are beginning to transform the industry [23].

Due to the increasing number of new entrants in the market, Original Equipment Manufacturers (OEMs) are no longer alone, and have to align their strategies based on what the competition offers, which provides customer-centric mobility and substantially interferes in the market [24–26].

Consequently, digital transformation changes the creation of value in companies, specifically in those where value is generated by physical elements, as is the case of the automotive industry [27]. The automotive industry is mainly being revolutionized by digital innovations, such as social networks, autonomous cars, connectivity, and big data [26,28], forcing them to adjust their business models to keep pace with technology, advances, and their effects [29–31], which are manifested, for example, through car sharing platforms or telematic services [18,32].

A dynamic capability is a learned and stable pattern of collective activity, through which the organization is capable of generating and constantly changing its operating routines, in the search for increased efficiency [33]. Dynamic Capabilities consist of detecting, capturing, and transforming microfoundations [4]. In this sense, with dynamic capabilities, the company can capture business opportunities, address threats, and create new opportunities, thus maintaining its competitiveness in the market [6]. In previous research, dynamic capabilities (DC) were classified into three types [7]:

(i) Sensing capability refers to the ability to diagnose the environment and understand the needs of the customers better than competitors; the ability to detect and shape opportunities and threats; the ability to seize these opportunities and the ability to maintain competitiveness by reconfiguring the organization's tangible and intangible assets [34–36]. By identifying potential qualified collaborating customers—lead users [37], firms operating in the automotive industry create a capability of detection—given that contact with customers at car dealerships enables a better understanding of their needs [38].

(ii) Seizing capability refers to the activity of addressing opportunities and threats [39]; the process in which substantial investments are devoted to address new opportunities and threats, which are encountered through sensing [34]. More specifically, supported by empirical findings, it is argued that this can be through the introduction of new products and services [37], as well as making incremental changes to existing business models [35]. Firms in the automotive industry acquire and assimilate external information and record it as part of the company's knowledge base to improve processes and products [40,41].

(iii) Innovation capability: describes the transformation process as the ability to configure organizational assets for the purpose of not becoming static and passive in the face of future changes [34]. The success of the product in the automotive industry is measured by the number of units sold per day and store. In addition to that, the definition and measurement of the number of units sold per store/day allow detecting product's commercial success [42]. In addition, in this industry, firms produce a limited number of products' units according to defined requirements [43]. Some previous articles have analyzed the contribution of dynamic capabilities to customer satisfaction [44–47], but in different industries rather than the automotive one.

A review of the literature aimed at identifying dynamic capabilities and indicators that measure these in the automotive sector has been carried out. The terms searched were: "Digital Transformation & Dynamic Capabilities", "Digital Transformation & Automotive Sector", and "Dynamic Capabilities & Automotive Sector". The most important databases used were: ABI Research, Econlit, Academic Search Premiere, Google Scholar, Springer, and Science Direct, from the period between years 2001 and 2020 [7].

The automotive industry is continuously impacted with the introduction of new technologies, which makes it necessary for organizations to adapt to the fast pace of growth [48]. Consequently, it is necessary to take into account the dynamic capabilities that these companies have, which also exceed core competencies, to be in continuous observation of changes in the environment and thus ensure the permanence of the industry in the market [49]. One of the most important results of deploying dynamic capabilities is the creation, renewal, and development of competencies and capabilities that allow the company to be constantly updated according to the changes occurring in the market [50]. The direct customer participation model in the automotive industry allows firms to increase the probability that the products offered under its own brands were more accepted and attractive than leading recognized brands [51]. Secondly, it also helps customers to perceive them differently, and achieve a positive welcome: "if you listen to your customers, it is easier to innovate successfully, with fewer risks; success is based on knowing how to connect" [52].

Customer satisfaction. The success of digital transformation will depend on creating customer value and understanding the need to improve processes and not just automate them [53]. In this sense, customer satisfaction through digital transformation is oriented to give them information regarding whether the chosen company is doing the right thing to respond to their demands [54].

In the automotive industry, thanks to the Internet of Things, artificial intelligence, and big data, new maintenance models have been developed, among which predictive maintenance stands out as an innovation for smart manufacturing, fault diagnosis, and assessment of the remaining lifetime of the vehicle [55].

The focus of digital transformation within an automotive company must be connected to the customer to improve their experience, either from the point of view of product quality or by improving connectivity [55]. In essence, companies capable of reducing costs, engaging customers, and making an efficient use of their assets with the implementation of digital technology will be among the winners of digital disruption [56].

In previous research of this group, dynamic capabilities-observed variables were assigned to the indicators from literature review and expert judgments. In addition, from a

quantitative methodology of exploratory analysis, the correct assignment of the indicators to each latent variable or dynamic capability was verified [7].

The research questions were focused on determining whether the generation of dynamic capabilities through digital transformation influences customer satisfaction in the automotive industry and its components (Table S1. Questionnaire Dynamic Capabilities).

Therefore, the main objective of this research was to verify if the dimensions of the dynamic capabilities (seizing, sensing, and innovation) can be grouped into a reliable construct for the automotive sector and assess the positive influence of dynamic capabilities on the creation of organizational value, with the consequent improvement in customer satisfaction.

Three partial objectives were raised in this paper; First, to build a construct of dynamic capabilities that incorporates the dimensions of seizing, sensing, and innovation. Second, to generate a factor with the variables of customer satisfaction. Finally, to quantify the influence of digital transformation in the building of dynamic capabilities on customer satisfaction in the automotive industry.

Research questions were evaluated by applying a mixed methods approach, combining qualitative and quantitative methodologies. For the approach of the theoretical model, a selection and assignment of the capabilities' indicators, the bibliography was used, and they were validated. Later, a structural equation model (SEM) was built to evaluate the influence of dynamic capabilities on customer satisfaction.

This work contributes to the grouping of the dimensions of the dynamic capabilities, which are internal to the organization, in a construct that was related to customer satisfaction (an external aspect to the organization) by considering that the digitalization issue is a clear priority. With the results of this research, we seek to help managers of organizations so that they can make a quick analysis of their market conditions, based on the effects of the development of dynamic capabilities on such a strategic dimension for any firm as it is on customer satisfaction. Hence, this research will validate the generation of dynamic capabilities through digital transformation in the automotive sector on customer satisfaction.

The article is structured as follows: after this introduction; where the theoretical framework to which the key concepts of digital transformation, dynamic capabilities, and these same concepts applied to the automotive industry are referred; in part 2, Material and Methods, the formulation of the hypotheses, the methodology, the sources of information, and the sample and the data collection instrument are described; in part 3, Results, the validation of the model is shown. Part 4 presents the discussion and conclusions that arise from verifying the results obtained in this research with the data provided by the literature review.

## 2. Materials and Methods

### 2.1. Hypothesis Approach

In previous work [7], a correct assignment of 15 observed variables to the dynamic capabilities' dimensions of sensing, seizing, and innovation was estimated (Table S1. Questionnaire of Dynamic Capabilities and Figure 1).

From the framework shown in Figure 1, in which the assignment of the observed variables to the dimensions of sensing, seizing, and innovation is made, the following research questions were presented: Could the dimensions of sensing, seizing, and innovation be grouped into a construct called "Dynamic Capabilities"? Did dynamic capabilities positively show influence on customer satisfaction? Therefore, two measurement models and a structural equation model were proposed (Figure 2).

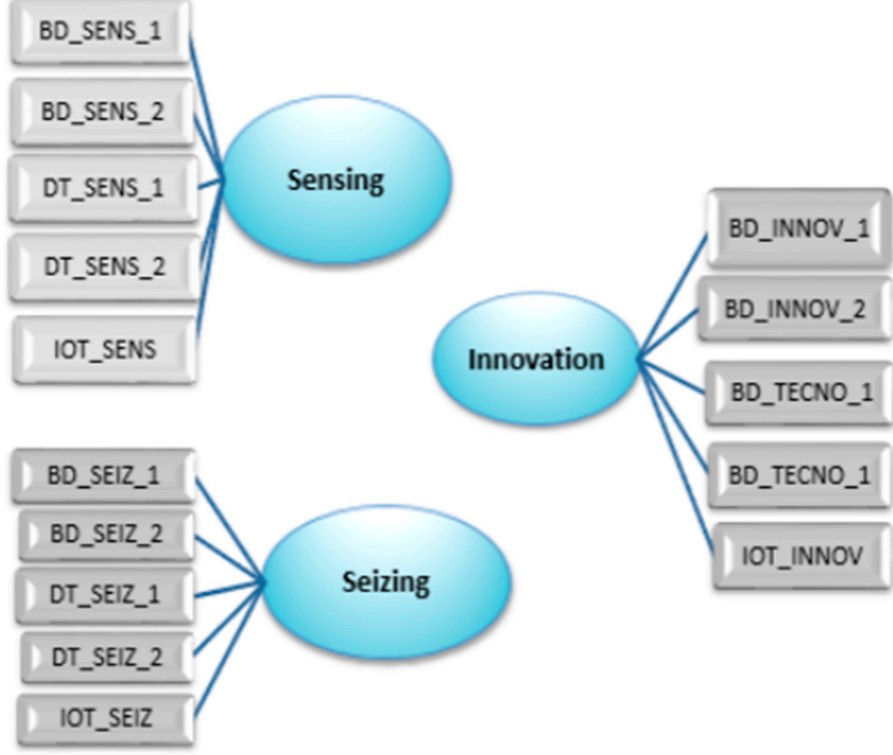

**Figure 1.** Indicators assigned to each latent variable.

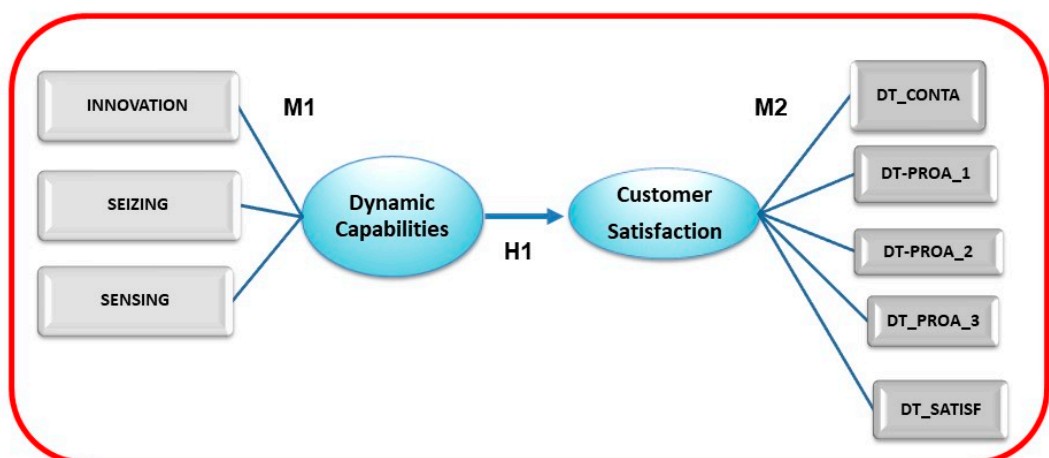

**Figure 2.** Models and hypotheses in theoretical model.

The proposed hypotheses were represented by means of a system of structural equations models in two stages. In the first stage, two reflective models (M1 and M2) were constructed. In a second stage, in Hypothesis 1, a structural equation model was built in which the relationships between dynamic capabilities (DynCap) and customer satisfaction (CustSatis) were evaluated.

Table 1 presents the inspiring literature review for the dynamic capabilities' indicators.

**Table 1.** Dynamic capabilities' indicators.

| Indicator | Authors |
|-----------|---------|
| Sensing | Teece, [34]; Helfat and Peteraf, [35]; Roy and Khokle [36]; Akram and Hilman [57]; Zhou et al., [58]; Bendig et al., [59]; Battisti and Deakins, [60]; Jacobi and Brenner [61]. |
| Seizing | Matysiak et al., [39]; Teece, [34]; Roy and Khokle [36]; Helfat and Peteraf [35]; Rigby et al., [62]; Kindström et al. [63]; Wang et al., [64]; Yeow, Soh and Hansen, [65]; Karimi and Walterm, [66] |
| Innovation | Helfat and Peteraf, [35]; Bendig et al., [59]; Kindström et al., [63]; Hodgkinson and Healey, [67]; Yeow, Soh and Hansen, [65]; Eisenhardt and Martin, [68]; Rotjanakorn, Sadangharn and Na-Nan, [6]; Teece, Pisano and Schuen, [69]. |

The applied questionnaire collected the following variables associated with customer satisfaction (Table S2. Review of satisfaction variables).

DT_CONTA. To what extent has digital transformation enabled us to identify the real needs of customers? Nowadays, traditional marketing methods are not sufficient to understand customer needs [70]. Consumer buying and selling behavior has rapidly evolved towards the use of mobile technology, online shopping, co-creation of value, among others, which has led to the development of new models for assessing the nature of consumer demand [71].

DT_PROA_1. To what extent the digital transformation has it enabled us to contact customers and solve the problems? Digital transformation has been focused on transforming customer experience, relationships, and processes [49]. This collaboration with the customer was the modern basis for innovation, as well as being an effective system to enable successful organizations to learn from the needs of their customers to meet their demands and improve performance. The success of digital transformation will depend on creating customer value and understanding the need to improve processes and not just automate them [53].

DT_PROA_2. To what extent the digital transformation has enabled us to be in direct contact with the customer by allowing us to collect data in order to offer products and additional services to the current ones anticipating your digital transformation needs? Customers are increasingly informed and connected, which allows them multiple alternatives of products and services [72]. In addition to liking the product, they must like the way it is being offered, which requires not only thinking about the product, but also thinking about the service [73].

DT_PROA_3. To what extent has the digital transformation made it possible to reduce vehicle accidents? Autonomous and assisted driving of vehicles will be made possible by the integration of advanced technologies, including GPS and sensors, cameras, connectivity, and algorithms [74]. The goal would be to make this type of driving available in less expensive vehicle models to help prevent accidents and save more lives [75].

DT_SATISF. To what extent has the installation of sensors, predictive models, and algorithm learning achieved more efficient driving? The vehicle is becoming an efficient machine that functions as a real time data transfer center [76]. These vehicles were designed to limit distractions and offer a personalized driving experience [77]. The system learns preferences passengers; in addition, it integrates with cell phones and offers coaching options, calendars, and navigation guidance [78].

**Hypothesis 1.** *Dynamic capabilities (DynCap) positively influence customer satisfaction (CustSatis) (Figure 2). This hypothesis is aimed at examining the direct effect of dynamic capabilities on firm results, using customer satisfaction as a proxy variable. The structural model for the relationship between dynamic capabilities and customer satisfaction is shown in Figure 2.*

### 2.2. Data Collection and Survey

In the classification of CNAE 29: Manufacture of motor vehicles, trailers, and semi-trailers; 1800 motor vehicle companies and 9060 automotive component companies appear in the annual detailed Enterprise statistics for industry [79]. The study population was composed by 106 automobile manufacturers and component manufacturers [3]. A random sample composed of 42 questionnaires in127 surveyed industries was collected during the period 2019–2020 in a pre-COVID-19 context. Incomplete surveys and those that showed logical inconsistencies were deleted. The sample size was calculated with a confidence of 95% (Z = 1.96), an unknown expected proportion ($p = 0.5$).

The survey included 28 items: 8 socioeconomic (age, gender, company size, professional profile, among others), 15 related to DC, and 5 related to customer satisfaction. The survey's reliability was verified through Cronbach's alpha, with values greater than 0.7, acceptable to confirm internal consistency. The complete survey showed a Cronbach's alpha of 0.93.

In Table S1, Questionnaire Dynamic Capabilities, a previous work where the 15 variables of dynamic capabilities were grouped in the dimensions of sensing, seizing, and innovation, is shown. Table 2 shows the high degree of association between the three dimensions considered in this analysis. Table 3 shows the statistical description of each indicator, showing the heterogeneity of the data and their degree of dispersion. In a previous work [7], the variables innovation, seizing, and sensing, as well as the SEM model relating the three dimensions of dynamic capabilities are described extensively.

**Table 2.** Correlations among latent variables.

| Correlation (*p*-Value) | Innovation | Seizing | Sensing |
|---|---|---|---|
| INNOVATION | - | 0.8051 (0.0000) | 0.6620 (0.0000) |
| SEIZING | | - | 0.7780 (0.0000) |

**Table 3.** Dynamic capabilities values for each latent variable.

| Variable | Scores (Median) | SD [1] | Minimum | Maximum | Q1 [2] | Q3 [3] |
|---|---|---|---|---|---|---|
| INNOVATION | 0.215 | 1.01214 | −2.522 | 1.996 | −0.654 | 0.643 |
| SEIZING | 0.173 | 1.01209 | −2.032 | 1.575 | −0.814 | 0.867 |
| SENSING | 0.311 | 1.01209 | −2.072 | 1.526 | −0.731 | 0.946 |

[1] Standard deviation, points, [2] First quartile, points, [3] Third quartile, points.

The customer satisfaction variables were evaluated by the Likert scale in this research. A Likert scale metric was used, from 1 (not important) to 5 (very important). In this case, the intervals between the points on the scale corresponded to empirical observations in the metric sense. A visual analog scale was displayed on each survey question presented to the interviewee.

Table 4 presents the statistical values for customer satisfaction according to each of the dynamic capabilities considered.

**Table 4.** Customer satisfaction values for each variable.

| Variable Observed | Scores (Mean) | SD [1] (CV [2] %) |
|---|---|---|
| DT_CONTA | 4.048 | 1.058 (26.14) |
| DT_PROA_1 | 3.857 | 1.117 (28.95) |
| DT_PROA_2 | 3.405 | 1.432 (42.07) |
| DT_PROA_3 | 3.786 | 1.423 (37.60) |
| DT_SATISF | 3.786 | 1.048 (27.66) |

[1] Standard deviation, points, [2] Coefficient of variation.

According to the results shown in Table 4, the five satisfaction variables showed high mean values and coefficients of variation greater than 25%. Cronbach's Alpha was greater than 0.7, and the questionnaire was validated for each of the indicators.

### 2.3. Statistical Analysis

The analyses were carried out in two stages. In the first phase, two reflective measurement models (M1 and M2) were used (Figure 2), which assessed the relationships between the constructs dynamic capabilities (DynCap) and customer satisfaction (CustSatis) and the indicators used. For this purpose, the internal consistency of each construct was measured (Cronbach's alpha and composite reliability); secondly, its convergent validity through the reliability of the indicator and the variance was extracted; and, finally, the discriminant validity between indicators and latent variables (Fornell Larcker criterion) and cross-loadings was found [80].

The causal relationships between dynamic capabilities (DynCap) and customer satisfaction (CustSatis) were measured in a second stage. To validate Hypothesis 1, a structural equation model (SEM) was developed (Figure 2). Both models were estimated using the partial least squares (PLS) procedure applying SmartPLS3 software [81], to test the relationships between indicators and latent constructs, as well as the hypothesized structural relationships between the latent constructs [82]. The criteria for choosing the algorithm were: the novelty of the phenomenon, its modeling is at an emerging stage, minimum PLS recommendations on sample size, prediction accuracy, and comparatively low demands on the multinormality requirements of the data were met [83].

Finally, bootstrapping was used to test the statistical significance of various PLS-SEM results such as path coefficients, Cronbach's alpha, HTMT, and $R^2$ values. In this research bootstrapping procedure was repeated until 5000 random samples were created [84].

### 3. Results

Table 5 shows the typology of the companies and the socio-demographic profile of the respondents. A total of 78.6% of the companies have more than 100 employees and 90% of the companies are consolidated with an age of more than 25 years, belonging to the automotive sector (71.4%) and with local and international activity (88%). The respondents were evenly distributed among staff, managers, and directors. The majority were men (97.6%) between 25 and 50 years of age (95.2%).

**Table 5.** Descriptive Data.

| Variable | Relative Frequency (%) |
| --- | --- |
| Number of employees | |
| Below 50 | 9.5 |
| Between 50 and 100 | 11.9 |
| Between 101 and 250 | 26.2 |
| More than 250 | 52.4 |
| Company age (y) | |
| below 25 | 9.5 |
| between 25 and 50 | 35.7 |
| more than 50 | 54.8 |
| Company position | |
| Staff | 31.0 |
| Middle manager/Manager | 38.1 |
| Executive/Director | 31.0 |
| Gender | |
| Male | 97.6 |
| Female | 2.4 |
| Age (y) | |
| between 25 and 50 | 95.2 |
| more than 50 | 4.8 |

**Table 5.** *Cont.*

| Variable | Relative Frequency (%) |
|---|---|
| Sector | |
|    Automotive | 71.4 |
|    Automotive components | 28.6 |
| Company operations | |
|    Spain (only) | 7.1 |
|    Europe (only) | 4.8 |
|    Global | 88.1 |

*Models and Hypothesis Assessment*

The model was estimated in two phases, firstly, the constructs used; and, secondly, the relationship between dynamic capabilities on customer satisfaction (Figures 1 and 3). All the capabilities presented in the model—sensing, seizing, and innovation—are shown in the Figure 1 of the introduction. In addition, an overview of the quality criteria is presented in Table 1 of methodology.

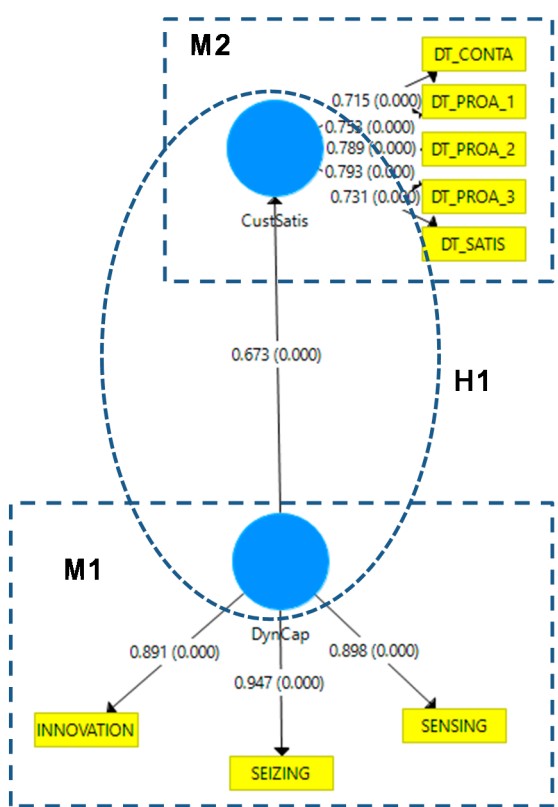

**Figure 3.** Models and hypothesis of the Dynamic Capabilities model on Customer Satisfaction.

Figure 3 and Tables 6–8 present the reflective and structural models, testing the hypotheses presented above. On the arrows of the model scheme, the coefficients are shown on a standardized scale from −1 to 1. Each construct was validated for its reliability and validity. Statistically significant relationships have *p*-values lower than 0.05. Dynamic capabilities showed a high impact on customer satisfaction (*p* = 0.000).

**Table 6.** Construct Reliability and Validity.

| | Cronbach's Alpha | Rho_A | Composite Reliability | AVE [1] |
|---|---|---|---|---|
| CustSatis | 0.814 | 0.822 | 0.870 | 0.573 |
| DynCap | 0.899 | 0.906 | 0.937 | 0.832 |

**Table 7.** Fornell Larcker Criterion.

|  | **CustSatis** | **DynCap** |
|---|---|---|
| CustSatis | 0.757 | |
| DynCap | 0.673 | 0.912 |

**Table 8.** Cross Loadings.

|  | **CustSatis** | **DynCap** |
|---|---|---|
| DT_CONTA | 0.715 | 0.379 |
| DT_PROA_1 | 0.753 | 0.527 |
| DT_PROA_2 | 0.789 | 0.492 |
| DT_PROA_3 | 0.793 | 0.590 |
| DT_SATIS | 0.731 | 0.520 |
| INNOVATION | 0.549 | 0.891 |
| SEIZING | 0.650 | 0.947 |
| SENSING | 0.637 | 0.898 |

In summary, the DC models' goodness-of-fit was adequate. Bootstrapping results are shown in Table 9. Confidence intervals assist in determining the significance of the relationships examined [85]. At a 95% confidence level, dynamic capabilities did impact customer satisfaction in the automobile industry, given the available data.

**Table 9.** Bootstrapping final results.

|  | **Sample** [1] | **SD** [2] | **T** [3] | ***p*-Value** | **Confidence Intervals** | |
|---|---|---|---|---|---|---|
|  |  |  |  |  | **2.5%** | **97.5%** |
| DT_CONTA <- CustSatis | 0.695 | 0.128 | 5.588 | 0.000 | 0.376 | 0.869 |
| DT_PROA_1 <- CustSatis | 0.748 | 0.096 | 7.847 | 0.000 | 0.501 | 0.886 |
| DT_PROA_2 <- CustSatis | 0.776 | 0.088 | 8.997 | 0.000 | 0.553 | 0.897 |
| DT_PROA_3 <- CustSatis | 0.792 | 0.077 | 10.295 | 0.000 | 0.537 | 0.888 |
| DT_SATIS <- CustSatis | 0.731 | 0.103 | 7.078 | 0.000 | 0.436 | 0.872 |
| INNOVATION <- DynCap | 0.890 | 0.034 | 25.940 | 0.000 | 0.793 | 0.937 |
| SEIZING <- DynCap | 0.946 | 0.019 | 50.598 | 0.000 | 0.897 | 0.973 |
| SENSING <- DynCap | 0.897 | 0.040 | 22.250 | 0.000 | 0.789 | 0.954 |

[1] Mean; [2] Standard deviation; [3] T Statistics.

The step-by-step results showed the following statistical indicators:

(a) From the parameters in Table 6, the reliability and validity of the two proposed constructs are accepted. Convergent validity was determined by the average variance extracted (AVE), defined as the mean value of the construct's indicators squared loadings. According to Fornell and Larcker, [86] the shared covariance is higher than the AVE for each of the two constructs. The resulting values (Table 6) show that the AVE values for customer satisfaction and dynamic capabilities were high (0.573 and 0.832, respectively) above the admitted value of 0.500 [87]. In terms of reliability, internal consistency reliability was assessed using Cronbach's alpha coefficient and composite reliability. Almost all measures exceeded the 0.700 threshold [88–90].

(b) Discriminant validity by Fornell and Larcker has been chosen as criteria for evaluating measurement scales that define latent constructs in our model (Table 7). All the correlations showed in Table 7 were greater than those obtained between the observed variables. Therefore, the indicators of both variables meet the required discriminant validity criteria [91].

In Table 8, the cross loadings of each indicator on latent variable are shown. It compares the cross-factor loadings of the indicators of a latent variable with the loadings of the other latent variables. As required, the factor loadings show higher values on its own than on than the others constructs.

Finally, the impact that dynamic capabilities had on the customer satisfaction was significant: 0.673 (path coefficient) and 0.0000 *p*-value. The structural equation model goodness-of-fit with a coefficient of determination R2 of 0.453, and size effect (f2) of 0.829. According to Cohen [92] an f2 greater than 0.35 is considered high.

## 4. Discussion

In this research, we evaluated how digital transformation has impacted through the deployment of dynamic capabilities, concretely sensing, seizing, and innovating, on customer satisfaction in the automotive industry [93]. These relationships among dynamic capabilities and customer satisfaction are clearly visible actions from the marketing perspective, as the dynamic capabilities are mainly happening inside the company, while the customer satisfaction is outside the company in the market place. Subsequently, both constructors were related through an SEM analysis. Our primary focus was to investigate whether dynamic capabilities of sensing, seizing, and innovation could be grouped to build a reliable indicator. In addition, dynamic capabilities were also examined to see how their deployment might increase customer satisfaction [94]. Therefore, the main interest of this work was the theoretical contribution to the development of the dynamic capabilities construct, such as the integration of innovation capabilities, sensing, and seizing, and the quantitative link of dynamic capabilities through digital transformation on customer satisfaction in the automotive sector.

### 4.1. Dynamic Capabilities Construct

Dynamic capabilities enable enterprises to develop the intangible assets to maintain processes in a sustainable performance [34]. Several researchers have focused on a double aspect; on the one hand, identifying the dimensions of dynamic capabilities, which disaggregated into the dimensions of sensing, seizing, and innovation, as reported by several authors such as: Bendig et al. [59], Roy and Khokle [36], Kevill et al. [95], Dixon et al. [96], and Martinez de Miguel, et al. [7]. On the other hand, dynamic capabilities in the automotive sector were widely described by Rotjanakorn, Sadangharn, and Na-Nan [6]; Leite, Borges, Dos Santos, Yutaka, and Castro [97]; Tondolo, Tondolo, Puffal, and Bittencourt [50]; Leite, [98]; Teece and Leih [99]; Camuffo and Volpato [100]; Christensen [101]; Alves [102]; Mesquita, Borges, Sugano, and Santos [103]; Lee [104]; Maynez, Valles, and Hernández [105]; Nakano, Akikawa, and Shimazu [106]; Makkonen, Pohjola, Olkkonen, and Koponen [107]; and Mamun, Muhammad, and Ismail [108].

This work is novel because the dimensions of sensing, seizing, and innovation in the automotive sector were grouped for the first time into a construct, which we have called Dynamic Capabilities. Similar models have been constructed by Mutmainah, et al. [109] in Higher Education (HE).

Subsequently, the dimensions of dynamic capabilities considered independently of each other were linked to the results, technological development, or innovation [6].

According to Lee and Yoo [110] sensing capability acts positively on seizing capability. Seizing directly influences the capability for innovation, because the new opportunities identified are used to create new products and services [111]. Consequently, the capability of sensing, positively influences the capability of seizing, as expressed by Lee and Yoo (2019), because the company can know the opportunities and needs of the environment and take advantage of this information to create new products and processes that will lead to the development of innovation [34]. Innovation activities are carried out with the purpose of favoring the survival and growth of the company, because a company that offers superior value to the competition, intervenes in the purchase intention and behavior of customers, resulting in best results [112]. The fact that, as reflected in the theory, companies obtain valuable information from this context and, in this way, they can learn about the needs of their customers and act accordingly [113]. The research presents some limitations that should be considered when contextualizing the work undertaken. The most representative one is the difficulty in obtaining a larger sample, because out of 142 surveys,

only 42 responses were obtained, due to the lack of vision on the usefulness of the study and the limited time respondents had to attend to the researcher, among other reasons. It is recommended to extend the sample to increase diversity and heterogeneity.

This measurement will help to develop active process improvement strategies to raise their market sensing, seizing, and innovation capabilities, and in this way, improve their managerial performance and seek a better positioning in the sector.

A future line of research would be to extend this study to other types of companies in different sector, in order to be able to measure the success of the dimensions selected grouped into the construct of DC.

### 4.2. Customer Satisfaction Construct

The customer satisfaction construct showed a high relationship with the five indicators of satisfaction considered. These results showed how the market is changing in a bidirectional way, to the extent that digital transformation has enabled the companies to identify the real needs of customers, contact them, and solve their problems [49]. On the one hand, customers are increasingly demanding more information and are looking for products adapted to their demand [114]. On the other hand, the market increases its diversity and offers them multiple options from which they can choose [115].

In view of the fact that the market offers them multiple options from which they can choose, they will demand personalized attention, quality, and novelty in terms of products and services [116]; consequently, as expressed by Stark [117], so that companies manage to adapt to the needs of their customers, they must offer innovative, quality, and environmentally friendly products. Consumers know that any company can satisfy their tastes and preferences, and this is something that every company that wants to have a differentiation must understand [118]. Companies that listen to their customers' needs and understand them hold the key to the development of new products and services [119].

Dynamic capabilities support new strategic designs that contribute to improve the viability and the sustainability of the automotive sector; the increasing pace of digital technology development also affects and brings major changes to all industries [116]. In addition, the automotive sector is heading from traditional engines to electrification with a clear focus on sustainability. The emergence of digital innovations is accelerating and intervening existing business models by delivering opportunities for new services. In this case, the automotive sector is leading trends such as car sharing, connectivity, and self-driving, creating new business models. Therefore, the capabilities that are generating increased added value could promptly develop a sustainable competitive advantage.

Customer satisfaction through digital transformation is oriented (connected to customer, improve their experience, and influence their purchase decision, mainly) to give them information regarding whether the chosen company is doing the right thing to respond to their demands [14]. The focus of digital transformation within an automotive company must be to be connected to the customer to improve their experience, either from the point of view of product quality or by improving connectivity [54].

In the case of the automotive industry, the information of the environment through technological tools is obtained from big data [120], the automation of sales forces using technological resources such as cell phones and tablets to maintain direct contact with customers [121], use of social networks, and the use of sensors in vehicles for autonomous driving and accident prevention [122].

This research, with its models and hypotheses, is focused on increasing customer knowledge and making processes more efficient, for which quantitative models are provided. Digitalization transforms the nature of products and the value creation process, so that companies can make unlimited combinations of products and services and thus integrate customer preferences into the joint creation of value [53]. Therefore, in companies that go hand in hand with digital transformation, the available technological resources, such as technical equipment, data storage devices, software, communication networks, among others, are used to provide customer service [123]. In this regard, authors such

as Lucas, Agarwal, Clemons, El Sawy and Weber [32], and Yoo, [124] have reported that digital technologies offer more flexible environments to create new organizational forms with customers, and as expressed by Hildebrant, Hanetl, Firk, and Kolbe [125] vehicle OEMs that have heterogeneous knowledge of digital technologies, and can integrate them into their companies and commercialize this knowledge, are better prepared to face the digital transformation.

Companies know that it is important to have the initiative to know what customers' needs are, and what opportunities they have to satisfy them [126]. Once the organization has detected the customer's need and the opportunities offered by the environment, managers focus on developing skills to exploit the potential of the opportunities detected and use them in the development of new products, processes, business, and services [127].

This measurement will help to develop active process improvement strategies to raise their customer satisfaction and assess the customers, and in this way, improve their managerial performance and seek better results in the sector.

A future line of research would be to extend this study to other types of companies, to be able to measure the customer satisfaction based on their dynamic capabilities.

### 4.3. Effect of Dynamic Capabilities on Customer Satisfaction

There is a lack of research examining customer satisfaction in the context of digital transformation, and we found an insufficient number of papers that have investigated the link between customer satisfaction and dynamic capabilities in the automotive sector [128]; even though, massive investments have been made in digital transformation and technology acceleration by both global and domestic IT companies [129].

In this paper, Hypothesis 1 was accepted, in which dynamic capabilities contribute positively to customer satisfaction in the automotive industry. Knowing this quantitative relationship through SEM is of great value to the company, as an improvement in dynamic capabilities contributes to an increase in customer satisfaction [99,130].

The dynamic capabilities selected are essential to promote creativity, and when strong, they make any firm able to cope with the uncertainty of innovation and competition [6]. Consequently, it is necessary to take into account the dynamic capabilities that these companies have, which also exceed the core competencies, to be in continuous observation of the changes in the environment and thus ensure the viability of the firm.

The digital transformation brings benefits for the automotive industry, among which the following can be highlighted: (a) improvements for the products adapted to customers demand; (b) development of new offers to multiple options from which customers can choose; (c) change in commercial strategies to sell a product, this time focusing on the customer experience; and (d) personalized attention, quality in terms of products and services. One of the greatest benefits that digital transformation brings to companies is the number of channels of interaction with customers, which allows them to obtain the necessary information about their requirements, preferences, and experiences [24].

Customers can access information from any device with internet access, and in any language, which allows them to compare quality attributes, prices, and recommendations from other users or customers [114]. In this sense, customer satisfaction through digital transformation is oriented to give them information regarding whether the chosen company is doing the right thing to respond to their demands [131].

### 5. Conclusions

This research opens new paths for knowledge regarding the automotive sector. Sensing, seizing, and innovation dimensions were grouped in a reliable indicator called "Dynamic Capabilities". The relationship between the dynamic capabilities construct and customer satisfaction by the SEM modeling was the main finding of this research.

In compliance with each specific objective set out in this research, the indicators used to determine each of the dynamic capabilities were suitable. In addition, the indicators used to determine the influence of sensing, seizing, and innovation capability have been adequate

for this purpose. In addition, the five indicators proposed to determine customer satisfaction through digital transformation were also suitable. Finally, the relationship between the dynamic capabilities and their effects on customer satisfaction has been quantified.

Of all the dynamic capabilities evaluated, the one that has the greatest influence on customer satisfaction is the capability of sensing, at least in this study, which could be explained because the company, having implemented the technological tools that allow closer contact with the customer, can detect what their needs and priorities are, which is the first step in making decisions that will give rise to business objectives, and from there, make the customer feel cared for, and taken into account for the decisions carried out by the organization.

Future research could delve into the relationship between the construct of dynamic capabilities with company results and the acquisition of competitive advantages by digital transformation within the automotive sector.

**Supplementary Materials:** The following supporting information can be downloaded at: https://www. mdpi.com/article/10.3390/su14084772/s1, Table S1. Questionnaire Dynamic Capabilities. Table S2. Review of satisfaction variables. Refs [132–137] has been cited in Table S2.

**Author Contributions:** Conceptualization and methodology, all authors. Formal analysis, software, data curation, and data processing, P.M.d.M.; statistical analysis, J.L.M. and A.G.; validation and investigation, J.L.M., C.D.-P.-H. and A.G.; supervision, A.G., C.D.-P.-H. and P.M.d.M.; project administration, C.D.-P.-H. and A.G.; data acquisition, P.M.d.M. and A.G. All authors have been involved in developing, writing, commenting, editing, and reviewing the manuscript. All authors have read and agreed to the published version of the manuscript.

**Funding:** This the authors received no financial support for the research, authorship, and/or publication of this article.

**Institutional Review Board Statement:** Not applicable.

**Informed Consent Statement:** Not applicable.

**Data Availability Statement:** This is not applicable as the data are not in any data repository of public access, however if editorial committee needs access, we will happily provide them, please use this email: pa1gamaa@uco.es.

**Acknowledgments:** We want to thank the OpenInnova High Performance Research Group at Rey Juan Carlos University and ECONGEST AGR267 Group at Córdoba University for the support to this research.

**Conflicts of Interest:** The authors declare no conflict of interest.

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
