# Peer review of "Impact of Dynamic Capabilities on Customer Satisfaction through Digital Transformation in the Automotive Sector"

_sustainability, doi:10.3390/su14084772_

Round 1

Reviewer 1 Report

Interesting work, with a good research part. Unfortunately, not everything is clear in the article.

  1. The Authors write about Dynamic Capabilities, however, there are not sufficiently scientifically reasoned to determine what competitive factors represent dynamic opportunities in the industry or market in general. The Authors should present a definition of dynamic capabilities, the achievements of literature in this area and explain why the elements indicated by them were adopted as the so-called dynamic capabilitties in automotoive sector.
  2. There are too many generalizations in hypothesis 1 that should not allow the conclusions drawn by the authors. The authors write, inter alia, that hypothesis 1 aimed to examine the direct effect of dynamics to develop a sustainable competitive advantage, using customer satisfaction as a proxy variable. Competitive advantage is very clearly defined and customer satisfaction is not a measure of competitive advantage. Therefore, it cannot be concluded that if something affects customer satisfaction, it also affects the competitive advantage. At best, this can have an impact on the company's competitiveness, in other words its ability to compete.
  3. The conclusions go well beyond the scope of the research.

Author Response

Reviewer 1

Answers to reviewer 1:

Thank you for the help. Your comments have been very helpful to improve the manuscript.

Review:

Interesting work, with a good research part. Unfortunately, not everything is clear in the article.

  1. The Authors write about Dynamic Capabilities, however, there are not sufficiently scientifically reasoned to determine what competitive factors represent dynamic opportunities in the industry or market in general. The Authors should present a definition of dynamic capabilities, the achievements of literature in this area and explain why the elements indicated by them were adopted as the so-called dynamic capabilities in automotive sector.
  2. There are too many generalizations in hypothesis 1 that should not allow the conclusions drawn by the authors. The authors write, inter alia, that hypothesis 1 aimed to examine the direct effect of dynamics to develop a sustainable competitive advantage, using customer satisfaction as a proxy variable. Competitive advantage is very clearly defined and customer satisfaction is not a measure of competitive advantage. Therefore, it cannot be concluded that if something affects customer satisfaction, it also affects the competitive advantage. At best, this can have an impact on the company's competitiveness, in other words its ability to compete.

The conclusions go well beyond the scope of the research.

  1. A definition of dynamic capabilities is included. Some main achievements of literature in this area are included and the way dynamic capabilities have been adopted in automotive sector is described

  1. You are right. We have removed the concept of competitive advantage in this hypothesis 1, as it is referred to the relationship between dynamic capabilities and customer satisfaction.

A great effort has been made to improve the manuscript according to the reviewer's suggestions. It was not easy, but we believe that the paper has been substantially improved now. All the different remarks offered by the reviewer have also been included.

Reviewer 2 Report

                                                                                                                                                                                                                           20.3.2022

Journal: Sustainability

Manuscript ID: sustainability – 1645578

Dynamic Capabilities on Custom Satisfaction in the automotive sector

Review:

The article in hand is related to Dynamic capabilities, Customer satisfaction, and structural equations modeling (SEM) in the automotive industry. The authors build and quantify the models of dynamic capabilities (M2), and they build and quantify the model (M1) of customer satisfaction. Then they build the SEM model combining these two models together. This is remarkable as the dynamic capability is something, which is mainly happening inside the company. And the actions to create these properties happen within the company. Customer satisfaction is mainly happening out the company in the market place. All the activities influencing on customer satisfaction are clearly visible actions by the marketing. The dynamic capabilities can have only indirect influence on the marketing. I consider it remarkable that the authors have built the connections between these two phenomena.

The objectives the authors state to be objectives of the article is to construct the SEM models (M2 and M1) and quantify the influence of the digital transformation in the influence of digital transformation in building of dynamic capabilities on customer satisfaction in automotive industry. Digital transformation has a significant role in this research even though it does not appear in the title nor in the figures of the article, but it appears in the text.

The definitions of dynamic capabilities and customer satisfaction are known in literature to which the authors are referring widely. The modeling and quantifying bring the digitality to the center of the paper, because many of the key variables that the models are showing up, include digital transformation related variable and aspects. And they seem to be in the center of the studied issues.  

I consider the SEM modeling as the main contribution of the article. The paper has been carefully and exactly written. In some places the technical details have become overwhelming and the issues are lost under the details in reporting.

The authors express one hypothesis in their text in the article: “Dynamic capabilities  (DynCap) positively influence on Customer satisfaction (CusSatis) .” Then the authors write: “This hypothesis aimed to examine the direct effect of the dynamics to develop a sustainable competitive advantage using customer satisfaction as a proxy variable.”  I think that the authors do not need the competitive advantage concept in this article. This concept includes many other aspects which are not in  the scope of this article. The competitive advantage  could  be deleted from this article without any loss in the content. The competitiveness as such is related to customer satisfaction but it is a long and wide separate story.

The aims in the article are well achieved. The role of digital transformation comes to the results as a new element without having basis in the theoretical introduction. The authors correctly note that digitalization with its many channels between the producing companies and the customers offer many benefits for

  • Customer communication back and forth and
  • A resource for building new properties between product and service combinations
  • A resource and tool to build totally new aspects for the existing and new products.

This all is to build up customer satisfaction upon customers´ needs.

Major improvement proposal:

I should propose the authors to take the digitalization issue already up in the theoretical part of the article as well in the introduction. Then the treatment in the empirical part is more appropriate. Accordingly the results for this issue deserve a better discussion in the conclusion part.  Perhaps you should include the world digital or digitalization also in the title.

Minor improvement proposals:

  1. In the Results section the authors should open better the achieved results. All the numerical indicators in the text should be referred in the text in the way that the reader can immediately check the given indicators in the tables. This is not now the case. If this is not always available or possible, the authors should give a clarifying explanation.
  2. Even in presenting technical / statistical results the story is a most important part. This means that the indicators have to be introduced, explained and interpreted in a way that the reader follows the thinking of the authors. It is not enough to tell that the indicator has some numerical value. It is also necessary to remind the reader at least once in the article what does the indicator tell and how it has to been calculated. I mentioned already about interpretation. This is the case here e.g. with indicators like Cronbach´s alpha, Rho_A, Composite reliability, AVE, Discriminant validity, discriminant reliability vs. correlation, facror loadings, path coefficient.

I repeat the story is important. You may even drop some indicators if they are overlapping. Use only the indicators which you really need to make the text more readable.

All in all, a good and carefully written article. Try to make it less technical, try to bring the finding better in the front.

Author Response

Answers to reviewer 2:

Thank you for the help. Your comments have been very helpful to improve the manuscript.

A great effort has been made to improve the manuscript according to the reviewer's suggestions. It was not easy, but we believe that the paper has been substantially improved now. All the different remarks offered by the reviewer have also been included

Review:

The article in hand is related to Dynamic capabilities, Customer satisfaction, and structural equations modeling (SEM) in the automotive industry. The authors build and quantify the models of dynamic capabilities (M2), and they build and quantify the model (M1) of customer satisfaction. Then they build the SEM model combining these two models together. This is remarkable as the dynamic capability is something, which is mainly happening inside the company. And the actions to create these properties happen within the company. Customer satisfaction is mainly happening out the company in the market place. All the activities influencing on customer satisfaction are clearly visible actions by the marketing. The dynamic capabilities can have only indirect influence on the marketing. I consider it remarkable that the authors have built the connections between these two phenomena.

The objectives the authors state to be objectives of the article is to construct the SEM models (M2 and M1) and quantify the influence of the digital transformation in the influence of digital transformation in building of dynamic capabilities on customer satisfaction in automotive industry. Digital transformation has a significant role in this research even though it does not appear in the title nor in the figures of the article, but it appears in the text.

The definitions of dynamic capabilities and customer satisfaction are known in literature to which the authors are referring widely. The modeling and quantifying bring the digitality to the center of the paper, because many of the key variables that the models are showing up, include digital transformation related variable and aspects. And they seem to be in the center of the studied issues.  

I consider the SEM modeling as the main contribution of the article. The paper has been carefully and exactly written. In some places the technical details have become overwhelming and the issues are lost under the details in reporting.

The authors express one hypothesis in their text in the article: “Dynamic capabilities  (DynCap) positively influence on Customer satisfaction (CusSatis) .” Then the authors write: “This hypothesis aimed to examine the direct effect of the dynamics to develop a sustainable competitive advantage using customer satisfaction as a proxy variable.”  I think that the authors do not need the competitive advantage concept in this article. This concept includes many other aspects which are not in  the scope of this article. The competitive advantage  could  be deleted from this article without any loss in the content. The competitiveness as such is related to customer satisfaction but it is a long and wide separate story.

Answer to reviewer:

1) We find your comments interesting; we have incorporated in the text some of your reflections that clarify the findings of the work and its interest for the reader. Besides, you are right, the Digital transformation has a significant role in this research. Therefore, digital transformation word has been incorporated in the title and developed in the text.

2) You are right, the concept of competitive advantage has been deleted from this article, as the article refers to customer satisfaction as outcome which is clearly different to the concept of competitive advantage.

The aims in the article are well achieved. The role of digital transformation comes to the results as a new element without having basis in the theoretical introduction. The authors correctly note that digitalization with its many channels between the producing companies and the customers offer many benefits for

  • Customer communication back and forth and
  • A resource for building new properties between product and service combinations
  • A resource and tool to build totally new aspects for the existing and new products.

This all is to build up customer satisfaction upon customers´ needs.

Major improvement proposal:

I should propose the authors to take the digitalization issue already up in the theoretical part of the article as well in the introduction. Then the treatment in the empirical part is more appropriate. Accordingly the results for this issue deserve a better discussion in the conclusion part.  Perhaps you should include the world digital or digitalization also in the title.

A: The paper does not present a theoretical part differentiated from the introduction. This means that in the introduction, the digitalization issue is a clear priority. This is the reason why the digitalization issue is included in the introduction of the article as other theoretical issues.

For this reason, authors have stressed in blue all the different theoretical arguments provided in the introduction on the digitalization issue.

Minor improvement proposals:

  1. In the Results section the authors should open better the achieved results. All the numerical indicators in the text should be referred in the text in the way that the reader can immediately check the given indicators in the tables. This is not now the case. If this is not always available or possible, the authors should give a clarifying explanation.
  2. Even in presenting technical / statistical results the story is a most important part. This means that the indicators have to be introduced, explained and interpreted in a way that the reader follows the thinking of the authors. It is not enough to tell that the indicator has some numerical value. It is also necessary to remind the reader at least once in the article what does the indicator tell and how it has to been calculated. I mentioned already about interpretation. This is the case here e.g. with indicators like Cronbach´s alpha, Rho_A, Composite reliability, AVE, Discriminant validity, discriminant reliability vs. correlation, factor loadings, path coefficient.

I repeat the story is important. You may even drop some indicators if they are overlapping. Use only the indicators which you really need to make the text more readable.

All in all, a good and carefully written article. Try to make it less technical, try to bring the finding better in the front.

A: We have tried to make the article less technical and brought the finding better in front.

Besides, the statistical part of the results has been clarified and opened to make it more understandable by the reader. 

Reviewer 3 Report

[Comment 1] Novelty

[Subcomment 1a] Please elaborate more about the paper’s contribution, while clearly stating what has/has not been done in which specific literature for each part in Figure 2. I assume the authors only compare their study with Reference [7].

[Subcomment 1b] Did the authors propose something new compared with existing M1 and M2 models? If yes, please mention in detail while comparing with the specific studies.

[Subcomment 1c] I don't think the authors conducted the literature review appropriately. There are less observations on existing and related studies. Some of not considered yet references are:

- https://www.emerald.com/insight/content/doi/10.1108/00251741111151181/full/html

- https://www.emerald.com/insight/content/doi/10.1108/03090560910989957/full/html

- https://www.emerald.com/insight/content/doi/10.1108/JBIM-10-2014-0215/full/html

- https://www.sciencedirect.com/science/article/pii/S0965856416306450

The authors need to state their paper’s contribution clearly while addressing the latest findings in the research field. Such lack of literature review hinders the readers to understand the research progress in the studied field.

[Comment 2] Research methodology

[Subcomment 2a] (page 3) Please clearly show the number of found and screened references, e.g., following the PRISMA statement (http://www.prisma-statement.org/).

[Subcomment 2b] (page 5) Which reference is used to derive the factors for the customer satisfaction variable? Please include it into the manuscript, or state that the authors propose theirs. If the authors propose a new model, why don't they use or improve any existing model?

[Subcomment 2c] (page 6) Please show the number of questionnaires before and after the removal at the earlier part of Section 2.2.

[Comment 3] Models and data

[Subcomment 3a] Why the authors use this kind of classifications in the manuscript for the dynamic capabilities while there are other classifications, e.g., sensing, learning, integrating, and coordinating capabilities (Pavlou, P.A. and El Sawy, O.A. (2011), “Understanding the elusive black box of dynamic capabilities”, Decision Sciences, Vol. 42 No. 1, pp. 239-273.) with much more number of citations (1,303 citations based on google scholar).

(I could not find Reference [7] used in this study, and its number of citations.)

[Subcomment 3b] Please upload Table S1 as a supplementary material.

[Comment 4] Clarity and writing quality

[Subcomment 4a] (page 3) Please give example for the “innovation” part to allow easy differentiation with "seizing capability" type.

[Subcomment 4b] (page 4) Please add spaces when necessary, e.g., for “presentsdiscussion”.

Author Response

Answers to reviewer 3:

Thank you for the help. Your comments have been very helpful to improve the manuscript.

A great effort has been made to improve the manuscript according to the reviewer's suggestions. It was not easy, but we believe that the paper has been substantially improved now. All the different remarks offered by the reviewer have also been included

Review

[Comment 1] Novelty

[Subcomment 1a] Please elaborate more about the paper’s contribution, while clearly stating what has/has not been done in which specific literature for each part in Figure 2. I assume the authors only compare their study with Reference [7].

A: Authors take the metrics to measure dynamic capabilities in the automotive industry from a previous article recently published in ESIC Market Journal, as previous proposals of indicators to measure dynamic capabilities in the automotive industry have not been provided before in literature review. Authors do not compare this study with reference 7, they just cite reference 7 as the unique previous article for which this one is an evolution.

Besides the contribution of this paper has been clarified

Our primary focus was to investigate whether dynamic capabilities of sensing, seizing and innovation could be grouped to build a reliable indicator. Besides, the effect of Dynamic Capabilities on Customer Satisfaction was assessment by SEM model

[Subcomment 1b] Did the authors propose something new compared with existing M1 and M2 models? If yes, please mention in detail while comparing with the specific studies.

A: In this research (regarding M1 and M2), we evaluated how digital transformation has impacted through the deployment of dynamic capabilities, concretely sensing, seizing and innovating, on customer satisfaction in the automotive industry. These relationships among dynamic capabilities and customer satisfaction are clearly visible actions by the marketing, as the dynamic capabilities are mainly happening inside the company and the customer satisfaction is out the company in the marketplace.

This work is novel because the dimensions of sensing, seizing and innovation in the automotive sector were grouped for the first time into a construct that we have called Dynamic Capabilities. Similar models have been constructed by Mutmainah, et al., [97] in Higher Education (HE). Traditionally, the dimensions of dynamic capabilities were considered independently of each other were linked to the results, technological development, or innovation.

[Subcomment 1c] I don't think the authors conducted the literature review appropriately. There are less observations on existing and related studies. Some of not considered yet references are:

- https://www.emerald.com/insight/content/doi/10.1108/00251741111151181/full/html

- https://www.emerald.com/insight/content/doi/10.1108/03090560910989957/full/html

- https://www.emerald.com/insight/content/doi/10.1108/JBIM-10-2014-0215/full/html

- https://www.sciencedirect.com/science/article/pii/S0965856416306450

The authors need to state their paper’s contribution clearly while addressing the latest findings in the research field. Such lack of literature review hinders the readers to understand the research progress in the studied field.

  1. The literature review has been revised and these references and other ones have been included

[Comment 2] Research methodology

[Subcomment 2a] (page 3) Please clearly show the number of found and screened references, e.g., following the PRISMA statement (http://www.prisma-statement.org/).

[Subcomment 2b] (page 5) Which reference is used to derive the factors for the customer satisfaction variable? Please include it into the manuscript, or state that the authors propose theirs. If the authors propose a new model, why don't they use or improve any existing model?

[Subcomment 2c] (page 6) Please show the number of questionnaires before and after the removal at the earlier part of Section 2.2.

 [Comment 3] Models and data

[Subcomment 3a] Why the authors use this kind of classifications in the manuscript for the dynamic capabilities while there are other classifications, e.g., sensing, learning, integrating, and coordinating capabilities (Pavlou, P.A. and El Sawy, O.A. (2011), “Understanding the elusive black box of dynamic capabilities”, Decision Sciences, Vol. 42 No. 1, pp. 239-273.) with much more number of citations (1,303 citations based on google scholar).

A: there are different classifications of dynamic capabilities. In this paper authors are focused in seizing, sensing and innovating capabilities as they take a previous model that they have built justifying the importance of these three capabilities in the automotive industry (reference 7)

Also, in previous research in other sectors, tourism, entrepreneurship, education, etc. we have used similar classification of dynamic capabilities

(I could not find Reference [7] used in this study, and its number of citations.).

  1. This article has been accepted to be published in brief in ESIC Market.

[Subcomment 3b] Please upload Table S1 as a supplementary material.

A: We have already uploaded it at the end of the article according to the editor's instructions

[Comment 4] Clarity and writing quality

A: writing clarity has been included.

[Subcomment 4a] (page 3) Please give example for the “innovation” part to allow easy differentiation with "seizing capability" type.

A: Example of the innovation part has been given to allow easy differentiation from seizing capability type

[Subcomment 4b] (page 4) Please add spaces when necessary, e.g., for “presentsdiscussion”.

A: This has been corrected.

Reviewer 4 Report

This is an interesting study, and the authors have valued the roles that Dynamic Capabilities play through sensing, seizing, and innovation through digital transformation into the automotive sector on customer satisfaction. In my opinion, the paper has some shortcomings in several areas that deserve improvement. Among the areas that need improvement works to provide better structure and organization. Some parts need to be clarified and rewritten. Specific examples are given below. 

The abstract is too long, accordingly to the journal  Instructions for Authors, it should be about 200 words maximum. 

In the introduction, there is a lack of carefully reviewing the research's current state or citing the key publications; moreover, it appears to be poorly organized. From 116 to 121, it is not clear. From 171 to 178 is not clear.

 Tables 1, 2, 3, and 4 need to be commented on and presented more clearly. 

From 181 to 182, better clarify. 

From 376 to 386, it seems like a part relating to a review more adapt to the first part of the paper. 

Given these shortcomings, the manuscript requires major revisions.

Author Response

Answers to reviewer 4:

Thank you for the help. Your comments have been very helpful to improve the manuscript.

Review

This is an interesting study, and the authors have valued the roles that Dynamic Capabilities play through sensing, seizing, and innovation through digital transformation into the automotive sector on customer satisfaction. In my opinion, the paper has some shortcomings in several areas that deserve improvement. Among the areas that need improvement works to provide better structure and organization. Some parts need to be clarified and rewritten. Specific examples are given below. 

The abstract is too long, accordingly to the journal  Instructions for Authors, it should be about 200 words maximum. 

A: The abstract has been adjusted to the number of words recommended

In the introduction, there is a lack of carefully reviewing the research's current state or citing the key publications; moreover, it appears to be poorly organized. From 116 to 121, it is not clear. From 171 to 178 is not clear.

A: Key publications have been added and the information in the introduction has been re-organized.

Tables 1, 2, 3, and 4 need to be commented on and presented more clearly. 

A: Tables 1, 2, 3 and 4 have been commented on and presented more clearly

From 181 to 182, better clarify. 

A: From 181 to 182 have been better clarified

From 376 to 386, it seems like a part relating to a review more adapt to the first part of the paper. 

A: This part has been adapted to the first part of the paper

Given these shortcomings, the manuscript requires major revisions.

A: A great effort has been made to improve the manuscript according to the reviewer's suggestions. It was not easy, but we believe that the paper has been substantially improved now. All the different remarks offered by the reviewer have also been included

Round 2

Reviewer 3 Report

[Comment 1] Novelty

Please add this novelty statement in the introduction section: “This work is novel because the dimensions of sensing, seizing and innovation in the automotive sector were grouped for the first time into a construct that we have called Dynamic Capabilities. Similar models have been constructed by Mutmainah, et al., [97] in Higher Education (HE). Traditionally, the dimensions of dynamic capabilities were considered independently of each other were linked to the results, technological development, or innovation.”

[Comment 2] Research methodology

The authors totally ignore my previous Subcomments 2a-2c, which actually might be important to ensure the completeness of the study. Not including such details hinders next researchers to conduct appropriate follow up researches. Although the authors prefer to ignore this revision request, I would not ask any further revision, considering other reviewers' comments.

[Comment 3] Reference

If you cannot find any active link for the reference published in ESIC Market, I suggest you remove the reference. It seems that the authors use this Reference [7] as the basis for capabilities importance assessment, but this reference is unsearchable, which actually might question the correctness of the used theory. If the authors could present any other study using this method, it would be helpful. Insisting on using Reference [7] only would cause confusion and doubt from the readers of this journal.

Author Response

We appreciate your effort. We have improved it in the different aspects that you have indicated. We now believe that it is suitable for publication. Thank you for helping us improve.

We apologize in advance as we clearly overlooked it. Hence, we procced as suggested to clarify each question.

Please add this novelty statement in the introduction section: “This work is novel because the dimensions of sensing, seizing and innovation in the automotive sector were grouped for the first time into a construct that we have called Dynamic Capabilities. Similar models have been constructed by Mutmainah, et al., [97] in Higher Education (HE). Traditionally, the dimensions of dynamic capabilities were considered independently of each other were linked to the results, technological development, or innovation.”

The paragraph is included in the discussion part, particularly, line number 224. On top of it, this idea has been reinforced in the introduction part. As suggested previously: “This work contributes to the grouping of the dimensions of the dynamic capabilities, which are internal to the organization, in a construct that was related to customer satisfaction (external aspect to the organization) by considering that the digitalization issue is a clear priority”.

[Comment 2] Research methodology

The authors totally ignore my previous Subcomments 2a-2c, which actually might be important to ensure the completeness of the study. Not including such details hinders next researchers to conduct appropriate follow up researches. Although the authors prefer to ignore this revision request, I would not ask any further revision, considering other reviewers' comments.

Subcomment 2a] (page 3) Please clearly show the number of found and screened references, e.g., following the PRISMA statement (http://www.prisma-statement.org/). [Subcomment 2b] (page 5) Which reference is used to derive the factors for the customer satisfaction variable? Please include it into the manuscript, or state that the authors propose theirs. If the authors propose a new model, why don't they use or improve any existing model?

We apologize in advance as we clearly overlooked it. Hence, we procced as suggested to clarify each question.

In the review of the dimensions of dynamic capabilities (sensing, seizing and innovation) more than 160 relevant references were used, widely collected in Martínez de Miguel, P., De-Pablos-Heredero, C. Montes-Botella, J.L. & Garcia, A. (2022). Review of the measurement of Dynamic Capabilities: a proposal of indicators for the automotive industry. ESIC Market, 53(1), e283. https://doi.org/10.7200/esicm.53.283. We have also used these dimensions in other previously developed models.

Following your recommendation, we created a table containing results of the review of the satisfaction variables. Although, we had to reduce the number of authors due to other reviewer recommendation (we eliminated the references of supplementary material, etc.). Finally, we verified that in each satisfaction variable there was at least one relevant reference. As per requested, the complete table is as follows. We also proceed to incorporate it into complementary material.

Table S2. Review of satisfaction variables

DT_CONTA

To what extent has it enabled us to CONTACT customers and solve the problems of digital transformation?

Alegre, J., Kishor, S., & Lapiedra, R. [135]

Davenport, T. and Spanyi, A. [53]

Nonaka, I., & Takeuchi, H. [136]

Pil, F. K., & Holwelg, M. [137]

Prahalad, C., & Ramaswamy, V. [138]

DT_PROA (1,2 &3)

To what extent has it allowed us to be in direct contact with the customer by allowing us to collect data in order to OFFER PRODUCTS  and/or ADDITIONAL SERVICES to the current ones anticipating your digital transformation needs?

Estevez, J. [72]

Von Leipziga, T., Gampa, M., Manza, D, Schöttlea, K.,

Ohlhausena, P., Oosthuizenb, G., Palma, D., von Leipzigb, K.

[54]

To what extent has the digital transformation made it possible to REDUCE VEHICLE ACCIDENTS?

Newman, D. [75]

World Economic Forum [139]

Ibanez, J., Laugier, C., Yoder, J. and Thrun, S. [74]

To what extent has the installation of sensors, predictive models and algorithm learning achieved MORE EFFICIENT DRIVING?

Accenture Research Deck (s.f.) [140]

World Economic Forum [139]

DT_SATISF

To what extent has digital transformation enabled us to identify the REAL needs of customers?

Gillpatrick, T. [71]

Moeller, L., Hodson, N. and Sangin, M. [56]

Teixeira, T. [70]

References :

Alegre, J., Kishor, S. and Lapiedra, R. Knowledge management and the innovation performance in a high-tech SMEs industry. International Small Business Journal: Researching Entrepreneurship, 2013.31(4), 454-470. https://10.1177/0266242611417472

Davenport, T. and Spanyi, A. Digital Transformation Should Start With Customers. MIT Sloan Management Review. 2019, source: https://sloanreview.mit.edu/article/digital-transformation-should-start-with-customers/

Nonaka, I., and Takeuchi, H. The knowledge-creating company: How Japanese companies create the dynamics of innovation.  New York, NY. 2015. Oxford University Press.

Pil, F. K., and Holwelg, M. Exploring scale: the advantages of thinking small. MIT Sloan Management Review, 2003. 44(2), 33-39.

Prahalad, C., and Ramaswamy, V. The future of competition: Co-creating unique value with customers. Boston, MA. 2004. Harvard Business School Press. https://10.1108/10878570410699249

Estevez, J. Transformación digital para enganchar a un cliente infiel. Insights. 2017, Avaliable: https://www.ie.edu/insights/es/articulos/transformacion-digital-enganchar-cliente-infiel/

Von Leipziga, T., Gampa, M., Manza, D, Schöttlea, K., Ohlhausena, P., Oosthuizenb, G., Palma, D., and von Leipzigb, K. Initialising customer-orientated digital transformation in enterprises. Procedia. 2017, 8, 517-524. https://publikationen.reutlingen-university.de/frontdoor/deliver/index/docId/1393/file/1393.pdf

Newman, D. Top 6 Digital Transformation Trens in the Automotive Industry Automotive. Forbes. 2017 https://www.forbes.com/sites/danielnewman/2017/07/25/top-6-digital-transformation-trends-in-automotive/?sh=4ecc291c54e1

World Economic Forum. Digital Transformation of Industries Automotive Industry, 2016. Available: https://www.accenture.com/_acnmedia/accenture/conversion-assets/wef/pdf/accenture-automotive-industry.pdf

Ibanez, J., Laugier, C., Yoder, J. and Thrun, S. Autonomos Driving: Contex and State of the Art. Hanbook of Intellegent Vehicles. 2012, 1271-1310. https://doi.org/10.1007/978-0-85729-085-4_50

Accenture Research Deck. The digital transformation of the automotive sector: From manufacturers to providers of mobility, page 9, “Connected vehicle.”, s.f. Available: http://reports.weforum.org/digital-transformation/from-drivers-and-passengers-to-connected-travelers/

Gillpatrick, T. The Digital Transformation of Marketing: Impact on Marketing Practice & Markets. Economics. 2019, 7, 139-156. https://doi.org/10.2478/eoik-2019-0023

Moeller, L., Hodson, N. and Sangin, M. The Coming Wave of Digital. 2018. https://www.cambeywest.com/subscribe2/?p=SNB&f=clogin&btx_i=477795&btx_pub_id=2762&btx_m=29158&btx_p=91

Teixeira, T. Unlocking the Customer Value Chain, Currencybooks.com. 2009.New York.

[Comment 3] Reference

If you cannot find any active link for the reference published in ESIC Market, I suggest you remove the reference. It seems that the authors use this Reference [7] as the basis for capabilities importance assessment, but this reference is unsearchable, which actually might question the correctness of the used theory. If the authors could present any other study using this method, it would be helpful. Insisting on using Reference [7] only would cause confusion and doubt from the readers of this journal.

We apologize as this was a typo on the reference. We do confirm that the article is published and the direct access has been placed accordingly.

Martínez de Miguel, P., De-Pablos-Heredero, C. Montes-Botella, J.L. & García, A. (2022). Review of the measurement of Dynamic Capabilities: a proposal of indicators for the automotive industry. ESIC Market, 53(1), e283. https://doi.org/10.7200/esicm.53.283

Reviewer 4 Report

I have appreciated the effort authors have made in order to improve the manuscript according to my suggestions.

Author Response

We appreciate your effort. We have improved it in the different aspects that you have indicated. We now believe that it is suitable for publication. Thank you for helping us improve.